# Potato Tuberworm *Phthorimaea operculella* (Zeller) (Lepidoptera: Gelechioidea) Leaf Infestation Effects Performance of Conspecific Larvae on Harvested Tubers by Inducing Chemical Defenses

**DOI:** 10.3390/insects11090633

**Published:** 2020-09-15

**Authors:** Dingli Wang, Qiyun Wang, Xiao Sun, Yulin Gao, Jianqing Ding

**Affiliations:** 1State Key Laboratory of Crop Stress Adaptation and Improvement, School of Life Sciences, Henan University, Kaifeng 475004, Henan, China; wangdingli@163.com (D.W.); 13673535200@139.com (Q.W.); sunxiao2017@126.com (X.S.); 2State Key Laboratory for Biology of Plant Diseases and Insect Pests, Institute of Plant Protection, Chinese Academy of Agricultural Sciences, Beijing 100193, China; gaoyulin@caas.cn

**Keywords:** *Phthorimaea operculella*, induced defense, tuber-feeders, potato, glycoalkaloid

## Abstract

**Simple Summary:**

Aboveground herbivory can affect belowground herbivore performance by changing plant chemicals. However, it is not clear how leaf feeding affects tuber-feeder performance in tuber-plants. We evaluated the effect of foliar feeding of the potato tuberworm *P. operculella* on the performance of conspecific larvae feeding on harvested tubers and measured the phytochemical changes in leaves, roots, and tubers. We found that aboveground *P. operculella* leaf feeding negatively affected the performance of conspecific tuber-feeding larvae, likely due to the increased α-chaconine and glycoalkaloids in tubers, suggesting that plant chemicals were reallocated among different tissues, with greater changes in metabolic profiles in leaves and tubers compared with roots. Thus, aboveground feeding by *P. operculella* during the growing season can change tuber resistance against the potato tuberworm during the warehouse storage of tubers.

**Abstract:**

Conspecific aboveground and belowground herbivores can interact with each other, mediated by plant secondary chemicals; however, little attention has been paid to the interaction between leaf feeders and tuber-feeders. Here, we evaluated the effect of the foliar feeding of *P. operculella* larvae on the development of conspecific larvae feeding on harvested tubers by determining the nutrition and defense metabolites in the whole plant (leaf, root and tuber). We found that leaf feeding negatively affected tuber larval performance by increasing the female larval developmental time and reducing the male pupal weight. In addition, aboveground herbivory increased α-chaconine and glycoalkaloids in tubers and α-solanine in leaves, but decreased α-chaconine and glycoalkaloids in leaves. Aboveground herbivory also altered the levels of soluble sugar, soluble protein, starch, carbon (C), nitrogen (N), as well as the C:N ratio in both leaves and tubers. Aboveground *P. operculella* infestations could affect the performance of conspecific larvae feeding on harvested tubers by inducing glycoalkaloids in the host plant. Our findings indicate that field leaf herbivory should be considered when assessing the quality of potato tubers and their responses to pests during storage.

## 1. Introduction

Plant–insect interactions can alter organismal performance and population dynamics in both above- and belowground plant tissues [1,2,3,4,5]. In these interactive systems, herbivores feeding on different plant tissues can both positively and negatively affect each other through insect-induced plant chemical defenses [6,7], as plant metabolites may be systemically reallocated to roots and leaves [8,9]. Furthermore, the pattern or direction of such above- and belowground interactions (i.e., negative, positive or neutral) may be affected by many factors such as plant species or herbivore species [10,11]. While most previous studies have documented the linkage between leaf and root feeding, little attention has been paid to the interaction between leaf feeders and tuber-feeders in tuber-plants [12].

Many studies suggest that aboveground herbivory significantly affects root herbivores by changing primary and secondary plant compounds. For instance, aboveground *Manduca sexta* (Lepidoptera: Sphingidae) infestations could significantly increase the performance of root nematodes by increasing the carbon source in roots [13]. In addition, changes in the composition and quantity of metabolites (i.e., glucosinolates, terpenoids and root volatiles) induced by aboveground herbivores could reduce the performance of root feeders [7,14,15]. A recent study further suggested that the integration of the growth and defense patterns of leaves and roots responding to aboveground or belowground herbivory could help to better understand the plant defense strategy across the whole plant [16]. However, how leaf feeders affect the defensive allocation of the tuber-plant (i.e., potato), from a whole plant perspective, remains unknown.

When plants are subjected to herbivore attacks, secondary metabolites which include glucosinolates, phenols, tannins, glycoalkaloids and terpenoids significantly change in different plant compartments and then affect insect performance [17,18,19,20]. For example, increased terpenoids in leaves induced by the root feeder *Agriotes lineates* (Coleoptera: Elataridae) hindered the growth of the leaf-feeder *Spodoptera exigua* (Lepidoptera: Noctuidae) on cotton plants [7]. Above- and belowground herbivory decreased the survival of adult *Bikasha collaris* (Coleoptera: Chrysomelidae) by increased tannin levels [8]. Similarly, tuber feeding by *Tecia solanivora* (Povolný) (Lepidoptera: Gelechioidea) larvae reduced the larval mass of foliar *S. exigua* and *Spodoptera frugiperda* (Lepidoptera: Noctuidae) by increasing the accumulation of chlorogenic acid and glycoalkaloids in potato plants, *Solanum tuberosum* L. [12]. Furthermore, herbivory-induced changes in primary metabolites involved in the function, growth and life history of different plant organs are closely related to herbivore performance [21,22,23]. For instance, increased nitrogen (N) in the root induced by leaf feeding caterpillars was found to increase the performance of root-feeding nematodes in tobacco plants [13]. Thus, studies on primary and secondary metabolites against herbivorous insects are important for predicting insect population dynamics.

The interaction between potatoes, *S. tuberosum,* and potato tuberworms, *Phthorimaea operculella* (Zeller) (Lepidoptera: Gelechioidea), provides an ideal system for estimating the potential importance of the systemically induced defense in tuber-plants. The potato tuberworm is an oligophagous species that is an important invasive pest of potato crops in temperate and subtropical regions [24]. *Phthorimaea operculella* is native to South America and can act as both an above- and belowground herbivore on potato plants, and cause extensive damage to tubers during storage [25]. Females lay eggs in leaves and soil around potato plants and on exposed tubers. The number of generations (2–13) varies geographically [25]. The potato is the fourth largest food crop in the world, and its tubers are rich in starch, protein and various vitamins. Potato plants contain two toxic glycoalkaloids, α-solanine and α-chaconine, in leaves, stems, petioles, roots, and tubers [26]. Several recent studies have assessed the resistance of tubers against the potato tuberworm to screen resistant varieties [27,28,29,30,31]. For instance, its female moths laid fewer eggs on hybrid varieties of wild and cultivated potatoes, and larval survival was negatively affected [32]. Recent studies have examined how tuber feeding by *T. solanivora* induced potato plant defenses that affected aboveground herbivore performance and nutrient transport to tubers [12,33]. However, the effect of leaf herbivory during the growing season is often ignored when assessing the level of tuber resistance to the potato tuberworm. Understanding the effect of leaf feeding during the growing season on tuber-feeder performance under storage conditions is important in predicting the level of tuber resistance against the potato tuberworm during warehouse storage.

The aim of this study is to investigate the effects of leaf herbivory on primary and secondary metabolites in leafs, roots and tubers, and the role of these chemicals in affecting tuber-feeders’ development performance in potato plants. Specifically, we ask: (1) how aboveground *P. operculella* infestations at the tuberization stage affect the subsequent performance of conspecific larvae feeding in harvestable tubers; (2) how aboveground herbivores affect plant growth and the reallocation of primary (soluble sugar, soluble protein, starch, C, N and C:N ratio) and secondary metabolites (α-chaconine, α-solanine and glycoalkaloids) among different plant parts (tubers, leaves, and roots). By addressing these questions, we aim to gain a better understanding of tuber-plant-induced defense in mediating the interactions of *P. operculella* larvae feeding leaves and tubers.

## 2. Materials and Methods

### 2.1. Plants

We used the potato cultivar FAVORITA in our experiments, which is widely planted in northern China. Potato seed-tubers were obtained from the Zhengzhou Vegetable Research Institute, Henan, China. Tubers were held in a plastic box (80 cm × 40 cm × 20 cm) until buds grew to about 1 cm in length, at which point tubers were cut into 25 g pieces, each with an eye bud. On March 2018, we planted tuber pieces 8 cm deep into pots (height: 25 cm, diameter: 30 cm) filled with a mixture of half field soil and half sphagnum peat moss (Jiangsu Beilei Technology Development Co., Ltd. Zhenjiang, Jiangsu, China). All plants were covered with fabric (nylon netting: 10 × 5 × 2 m) in an open-sided greenhouse with natural temperature and light at Henan University, Kaifeng, China (34°49′ N, 114°18′ E).

### 2.2. Insects

In 2017, we collected *P. operculella* larvae from a field in Xuanwei City (Yunnan Province, China) and reared them on fresh potato tubers at 27 ± 2 °C, 70–80% R.H. (Relative Humidity) and a 12:12 h L:D (Light: Dark) photoperiod in a nylon cage (40 cm × 40 cm × 40 cm) whose bottom was covered with sterilized sand, until the mature larvae left the tuber and pupated in the sterilized sand. After about 10–15 days, the adults began to emerge, then we placed 30 pairs in a plastic container (height: 25 cm, diameter: 20 cm) covered with gauze and a 10% honey solution. Containers were topped with filter papers for oviposition. Two days later, filter papers with eggs were collected and stored in zip-lock bags in the above laboratory conditions (27 ± 2 °C, 70–80% R.H. and a 12:12 h L:D photoperiod); larvae that hatched within 12 h were used in our experiment.

### 2.3. Leaf Herbivory by P. operculella Larvae and Plant Responses

To evaluate how potato plants responded to aboveground *P. operculella* larvae attacks, we conducted a greenhouse experiment. After five weeks of plant growth, we selected intact plants of similar sizes (height: 20 ± 1 cm) and established three herbivory levels of aboveground feeding by *P. operculella* larvae: (1) control (0 larvae per plant), (2) low herbivory (9 larvae per plant) and (3) high herbivory (18 larvae per plant), and each level had 20 replicates. We inoculated newly hatched larvae (within 12 h) on the third, fourth, and fifth plant leaves, counting from top to bottom, with a soft paintbrush, and covered the infected leaves with a small net bag (15 cm length and 10 cm width) to prevent larvae from moving and damaging other tissues (i.e., stems, tuber). Control plants were also covered with small net bags but were not infested. Sixty plants were individually caged with nylon nets (80 cm height and 35 cm diameter) to protect them from other pests, then were randomly placed in the greenhouse with natural temperature and light. After 20 days, we removed all the larvae carefully until all plants were harvested for analysis.

All plants were harvested in June 2018. We measured the height of half the plants, and then we cut off the aboveground parts with scissors. Underground tubers were carefully picked with scissors and the soil was removed with a brush and weighed. All the roots were collected and washed clean with water. The aboveground parts of the plant, tubers and roots were packed in paper bags and dried at 60 °C for four days and weighed. All samples were ground into powder with a ball mill (Heng’ao HMM-400A, Tianjin Heng’ao Technology Development Co., Ltd., Tianjin, China) for chemical analysis. For the other half, we only collected tubers for insect bioassay. 

### 2.4. Effects of Aboveground Herbivory on the Performance of Larvae Feeding on Tubers

To assess the effect of aboveground herbivory on the developmental performance of larvae feeding on tubers, we conducted laboratory tests with the harvested tubers obtained from plants with different herbivory levels. The tubers stored for three days were used in this bioassay experiment. We selected three tubers per plant, and thirty tubers were tested for each aboveground herbivory treatment. We inoculated a single neonate larva into a tuber using a soft paintbrush and then placed individual tubers into a transparent glass container (12 cm tall, 10 cm in diameter) with 2 cm of sterilized sand at the bottom of each container. After ten days, we collected pupae from the sand every day, weighed them and individually put them in small petri dishes (1 cm tall, 5 cm diameter) with tissue paper. When adults emerged, we determined their sexes using the methodology suggested by Rondon [24] and tracked the gender of their pupae and larvae. We also recorded the developmental time of larvae and pupae in the whole bioassay.

### 2.5. Analysis of Plant Chemistry

For plant chemistry, we measured the N contents, C contents and concentrations of soluble sugars, soluble protein and starch in tubers, leaves, and roots. The soluble sugar concentrations were determined by spectrophotometry (Thermo Scientific GENESYS 10S, Waltham, MA, USA) at 630 nm wavelength, according to Elleuch [34]. For the analysis of starch levels, the starch in tissue samples was broken down into soluble sugars with perchloric acid in boiling water conditions (100 °C) and the starch levels were then determined by spectrophotometry as per analysis of the soluble sugar concentrations. The soluble protein concentrations were determined according to Bradford [35]. We also determined the N and C contents with the elementar vario Macro CUBE (Hanau, Germany) analysis system.

To examine plant defensive chemicals, the concentrations of α-solanine and α-chaconine were determined in leaves, roots, and tubers. The two glycoalkaloids were extracted following methods reported by Sotelo and Serrano and Friedman et al. [36,37], and their levels were determined with high performance liquid chromatography (Agilent 1260, Palo Alto, CA, USA). The column was reversed-phase ZORBAX SB-C18 (4.6 × 250 mm, 5 µm). The mobile phase was 0.05 M monobasic ammonium phosphate buffer-acetonitrile (65:35, *v/v*), the flow rate was 1 mL/min, and the operating condition was room temperature (25 °C). The UV absorbance was measured at 210 nm and the injected sample size was 20 µL. Because the concentrations of α-solanine and α-chaconine together account for 95% of the total glycoalkaloid concentrations [38], we quantified the total glycoalkaloid concentrations by dividing the total concentration of α-solanine and α-chaconine by 0.95.

### 2.6. Data Analysis

To assess plant growth and plant chemical response to aboveground herbivory, we used one-way ANOVAs to examine the effects of larval treatments on aboveground biomass, root biomass, tuber biomass, total biomass, plant height, tuber weight and the levels of soluble sugar, soluble protein, starch, total C, total N, the C:N ratio, α-solanine, α-chaconine, and total glycoalkaloids in three plant tissues (roots, leaves, and tubers). To estimate the developmental fitness of conspecific larvae fed on harvestable tubers under different treatments, we used two-way ANCOVAs to analyze larval and pupal developmental times and pupal weight, with treatments and sex as main factors, and individual tuber weights as a covariate. Differences among treatments were determined using post hoc Tukey’s HSD (Honestly Significant Difference) test for multiple comparisons. The data satisfied the assumption of the homogeneity of variance. All experimental data were analyzed using the software R, version 3.4.2 (R Development Core Team, 2017) [39].

## 3. Results

### 3.1. Plant Growth Response to Aboveground Herbivory

Larval herbivory treatments affected plant growth. Control plants had significantly greater heights (F_2,27_ = 4.597, *p* = 0.019; Figure 1a) and aboveground biomass (F_2,27_ = 7.371, *p* = 0.003; Figure 1b) than plants exposed to high herbivory treatments. Additionally, plants in the two aboveground herbivory treatments had significantly lower root biomass than control plants (F_2,27_ = 14.676, *p* < 0.001; Figure 1c). The tuber biomass (F_2,27_ = 3.174, *p* = 0.058; Figure 1d) from control plants was greater than for the tubers from plants subjected to aboveground *P. operculella* herbivory, but these differences were not significant. The total biomass (F_2,27_ = 5.876, *p* = 0.008; Figure 1e) was significantly greater in control plants than those exposed to high herbivory treatments. However, tuber weight did not differ significantly among the three herbivory treatments (F_2,27_ = 2.431, *p* = 0.107; Figure 1f). 

### 3.2. Development Performance of Tuber-Feeding Larvae

Neither treatment (aboveground herbivory level) nor insect sex affected the larval developmental time, but the interaction of these factors did have a significant effect (Table 1). Female *P. operculella* larvae which fed on tubers from plants subjected to the aboveground high-herbivory level had significantly longer developmental times than larvae that fed on tubers from plants with no aboveground herbivory (F_2,32_ = 4.25, *p* = 0.023; Figure 2a). There was no significant effect of aboveground herbivory on the developmental time of male *P. operculella* larvae (F_2,31_ = 0.478, *p* = 0.625; Figure 2b). The female pupal weight was significantly greater than the male pupal weight (Table 1). However, treatments had no significant influence on the female pupal weight (F_2,32_ = 0.078 *p* = 0.925; Figure 2c). Male pupae from low-herbivory tubers had significantly lower weights than those from control tubers (F_2,31_ = 4.757; *p* = 0.016; Figure 2d). Furthermore, there was no significant difference between the developmental times of female and male *P. operculella* pupae which fed tubers (as larvae) from different treatments (female pupae: F_2, 32_ = 0.301, *p* = 0.742; Table 1, Figure 2e; male pupae: F_2,31_ = 2.611, *p* = 0.09; Table 1; Figure 2f).

### 3.3. Effects of Leaf Herbivory on Plant Nutrients

The soluble sugar levels of leaves from plants exposed to the aboveground low herbivory level was significantly greater than those from control plants (F_2,27_ = 3.345, *p* = 0.05; Figure 3a). However, aboveground herbivory significantly reduced leaf soluble protein concentrations by 19% (F_2,27_ = 11.843, *p* < 0.001; Figure 3b). The leaf starch concentrations were 15% lower in plants exposed to aboveground herbivory compared with control plants (F_2,27_ = 8.579, *p* = 0.001; Figure 3c). The root soluble sugar concentrations from plants with a high herbivory level were significantly lower than that from low herbivory and control plants (F_2,27_ = 3.643; *p* = 0.04; Figure 3d). However, there was no impact of aboveground herbivory on the soluble protein and starch concentrations in roots (Figure 3e,f). The high herbivory treatment caused a significant decrease in tuber soluble sugar concentrations compared to the other two treatments (F_2,27_ = 6.277, *p* = 0.006; Figure 3g). However, the concentrations of soluble protein (F_2,27_ = 10.049, *p* = 0.001; Figure 3h) in tubers from plants exposed to a high level of herbivory were significantly greater than those from both control plants and those exposed to low herbivory. In addition, compared to the low level of herbivory, high aboveground herbivory significantly increased starch concentrations in tubers (F_2,27_ = 8.912, *p* = 0.001; Figure 3i).

Aboveground herbivory caused a significant decrease in leaf C contents compared to control plants (F_2,27_ = 4.26, *p* = 0.025; Figure 4a). The leaf N content of plants with low level of herbivory was significantly lower than that of plants in the other treatments (F_2,27_ = 7.748, *p* = 0.002; Figure 4b), while the C:N ratio was significantly greater (F_2,27_ = 8.732, *p* = 0.001; Figure 4c). However, in roots, there was no impact of aboveground herbivory on the C and N contents, or the C:N ratio (Figure 4d–f). Tubers from plants exposed to a low level of herbivory had significantly lower C contents than that exposed to a high level of herbivory and control plants (F_2,27_ = 12.429, *p* < 0.001; Figure 4g). The tuber N content of plants exposed to aboveground low herbivory levels was significantly greater than that of control plants (F_2,27_ = 5.506, *p* = 0.01; Figure 4h). However, the low herbivory treatment significantly reduced the C:N ratio (F_2,27_ = 7.99, *p* = 0.002; Figure 4i) compared to the other treatments.

### 3.4. Effects of Leaf Herbivory on Plant Secondary Chemicals

The leaf α-solanine concentrations from plants exposed to aboveground herbivory were significantly greater than that of control plants (F_2,27_ = 9.153, *p* = 0.001; Figure 5a), while aboveground herbivory reduced leaf α-chaconine concentrations by 44% (F_2,27_ = 95.022, *p* < 0.001; Figure 5b). However, the leaf total glycoalkaloid concentrations were 28% lower in plants exposed to aboveground herbivory compared to control plants (F_2,27_ = 83.768, *p* < 0.001; Figure 5c).

There was no effect of aboveground herbivory on the levels of α-solanine, α-chaconine and glycoalkaloids in roots (Figure 5d–f). There was no change in tuber α-solanine concentrations among three leaf herbivory levels (Figure 5g). However, the concentrations of tuber α-chaconine from plants exposed to aboveground herbivory were 1.3 times that of control plants (F_2,27_ = 15.316, *p* < 0.001; Figure 5h). Furthermore, the concentrations of tuber total glycoalkaloids in the aboveground herbivory plants were 1.2 times greater than in control plants (F_2,27_ = 14.055, *p* < 0.001; Figure 5i).

## 4. Discussion

In this study, we found that aboveground infestation by larvae of *P. operculella* significantly affected the larval developmental time and pupal weight of conspecific larvae feeding on tubers after harvesting. In addition, our results suggest that aboveground herbivory systematically drove the reallocation of phytochemicals among leaves, roots, and tubers, whereby the source tissue (leaves) and sink tissue (tubers) showed greater variation in the levels of plant nutrients and plant defensive compounds than roots. Together, these results indicate that, in the field, aboveground herbivory potentially affects the subsequent performance of larvae of the same herbivore when they feed on stored tubers in warehouses, likely due to the altered metabolic profiles in tubers. Our findings improve our understanding of the importance of early-season aboveground herbivory in the field when predicting tuber responses to conspecific damage in warehouses.

A growing body of evidence shows that the accumulation of glycoalkaloids in potato tubers increases the risk of food toxicity to humans and may act as defensive metabolites against insects [40,41,42]. For instance, an increase in the levels of induced glycoalkaloids reduced the performance of both *S. exigua* and *S. frugiperda* which fed on potato leaves [12]. In this study, we found that leaf infestations by *P. operculella* negatively affected the development of conspecific larvae which fed on the tubers produced, likely due to the induced glycoalkaloid accumulation in the harvested tubers. However, it remains unclear if its performance was affected by the low glycoalkaloid (α-chaconine) concentrations in tubers because *P. operculella* can feed on potato leaves with higher glycoalkaloid concentrations [43]. It might be that the changes in the proportions of α-solanine and α-chaconine influenced the variation in the larval development of tuber-feeders. In our study, we only measured glycoalkaloids (i.e., α-solanine and α-chaconine) as an index of defensive resistance. In the future, other compounds in potatoes (i.e., phenolic compounds) should be examined to fully understand the mechanisms involved in the defense of tubers against herbivores.

Primary chemical compounds not only play an important role in plant physiological processes, but also affect the performance of insects [44]. In this study, nitrogen content significantly increased in the tubers when plants were exposed to leaf herbivory. A previous study suggested that relatively low amounts of nitrogen negatively affect insect performance [45]. By contrast, we found that an increase in the tuber nitrogen content, as induced by leaf herbivory, had unfavorable effects on tuber herbivores which was indicated by a longer larval developmental time and lower pupal weight. However, many herbivores also use soluble sugars and proteins efficiently to acquire nourishment [46]. In the current study, we also found that aboveground herbivory significantly affected tuber nutritional quality; for instance, the high aboveground herbivory treatment reduced tuber soluble sugar but increased tuber soluble protein. It is well documented that the fitness of herbivorous insects depends on host plant quality, physical traits, nutritional compounds, and secondary metabolites [47]. Therefore, in this study, both changes in tuber nutrients and defensive compounds affected the larval development of *P. operculella* in tubers.

In this study, we found that high herbivory (18 larvae treatment) significantly increased the larval developmental time of female larvae of *P. operculella* which fed on the tubers produced, but males were not affected. Similarly, a previous study showed that there was a significant difference in the development of female and male *P. operculella* larvae in tubers from different cultivars [48]. Interestingly, the significant interaction between sex and treatment on the larval developmental time influenced two different responses in females and males to aboveground herbivory. In addition, our study showed that aboveground herbivory reduced the soluble sugar and total carbon contents in the tubers. As Fenemore determined [49], sugar is responsible for the lifetime fecundity of the potato tuberworm; a decrease in the soluble sugar and total carbon contents should cause a lower fecundity in the potato tuberworm. Altogether, aboveground *P. operculella* infestation may negatively and differently affect the performances of female and male tuber-feeders by reallocating nutritional and defensive metabolic profiles in tubers.

When plants are subjected to herbivory, the reconfiguration of metabolism is triggered and this extends to the whole plant, imposing restrictions on the nutrition supply to the herbivore [50]. In our study, the metabolite profiles underwent dramatic changes after plants were subjected to foliar herbivory, with the largest changes recorded in leaf and tuber tissues, but little change in metabolite allocation (i.e., soluble sugar) in roots. This effect may be due to the source–sink relationship between tubers and plant foliage, in which nutrients are transported from leaves to tubers during tuberization [31]. The plant metabolite profiles may have been reconfigured mainly in both the leaf and tuber tissues when we subjected the potato plants to foliar herbivory. Our results suggest that either low or high herbivory by *P. operculella* affected the concentration of tuber chemicals (i.e., soluble sugar, soluble protein, starch, C, N, C:N ratio, α-chaconine, α-solanine and glycoalkaloids). Similarly, the chemical levels in the leaves were also affected, except for α-solanine. However, in roots, only high herbivory reduced soluble sugar levels. These significant changes in phytochemical reallocation between leaves and tubers in responses to aboveground herbivory may indicate an intimate connection between the plant–herbivore interaction and plant physiology.

## 5. Conclusions

In summary, we found that leaf *P. operculella* infestation during the growing season influenced the development of larvae feeding on harvested tubers by increasing the larval developmental time and reducing pupal weight. Our chemical analysis further suggests that increased defensive glycoalkaloids and changes in tuber nutrients caused by early-season leaf herbivory by *P. operculella* led to changes in the performance of conspecific tuber-feeders. Given that potato tuberworms are multivoltine, we recommend considering the effects of the induced defense from leaf herbivory during the growing season on tubers when assessing the harvested-tuber resistance against the potato tuberworm. Furthermore, because there are many other leaf herbivores on potatoes, their damage to leaves may also induce chemical change in tubers which can also affect the potato tuberworms in the warehouse. Thus, future field and lab studies should focus on how the potato plant, leaf herbivores and tuber herbivores interact to provide new insights into the management of potato pests.

## Figures and Tables

**Figure 1 insects-11-00633-f001:**
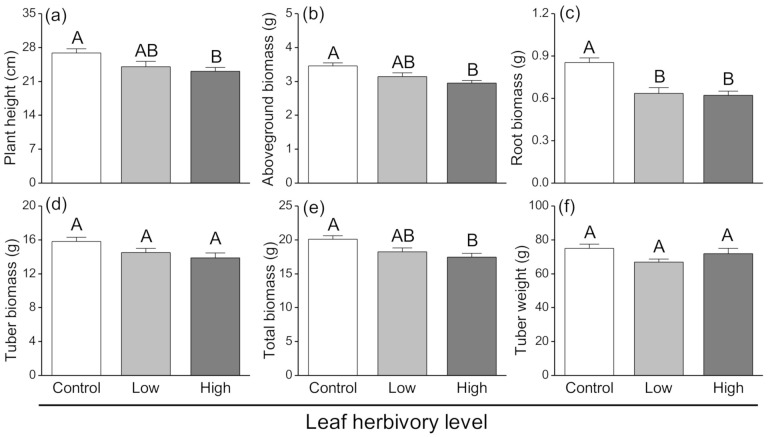
Effects of aboveground herbivory by *Phthorimaea operculella* with different levels on (**a**) plant height, (**b**) aboveground biomass, (**c**) root biomass, (**d**) tuber biomass, (**e**) total biomass and (**f**) tuber weight after harvesting all potato plants. Control (white bars) indicate no aboveground herbivory, Low (gray bars) indicate aboveground herbivory with nine larvae, High (black bars) indicate aboveground herbivory with eighteen larvae. Data are means ± SE (Standard Error). Different letters indicate significant differences among treatments (*p* < 0.05) based on post hoc Tukey’s HSD test.

**Figure 2 insects-11-00633-f002:**
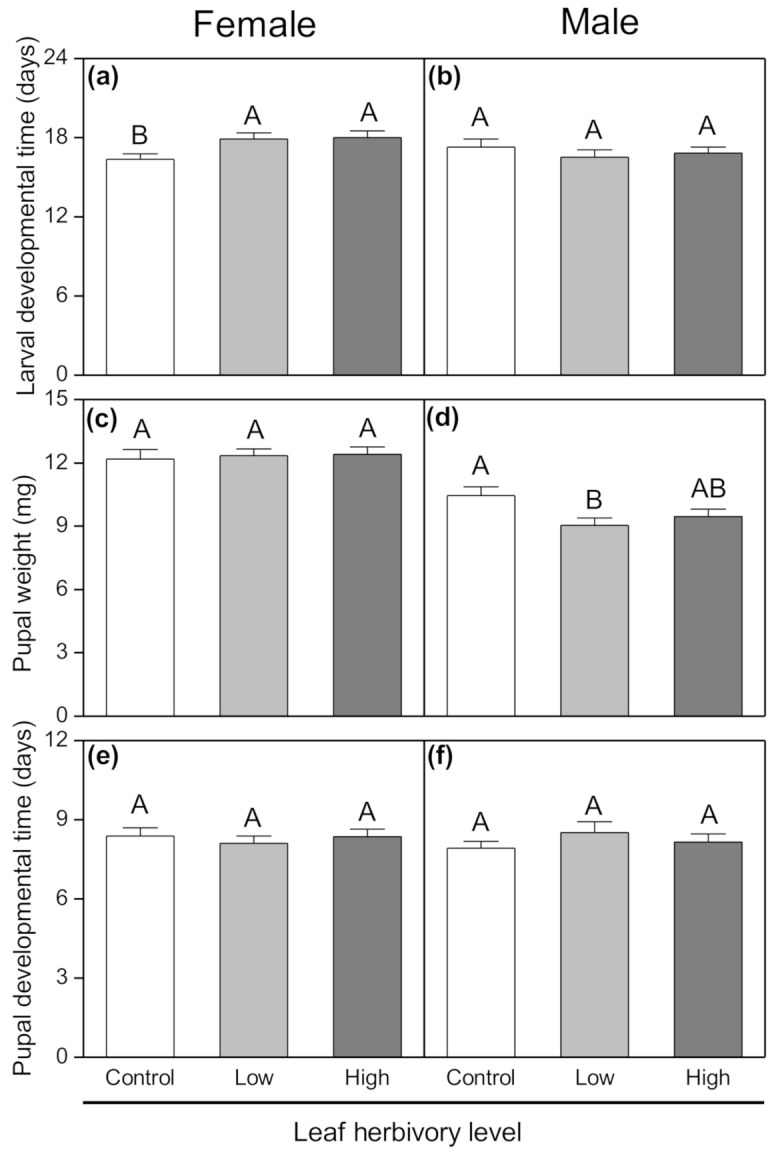
Changes in (**a**,**b**) larval developmental time, (**c**,**d**) pupal developmental time, and (**e**,**f**) pupal weight of female and male *Phthorimaea operculella* in tuber tissues for potato plants exposed to different herbivory levels of conspecific larvae. Data are means ± SE. Different letters indicate significant differences among treatments (*p* <0.05) based on post hoc Tukey’s HSD test.

**Figure 3 insects-11-00633-f003:**
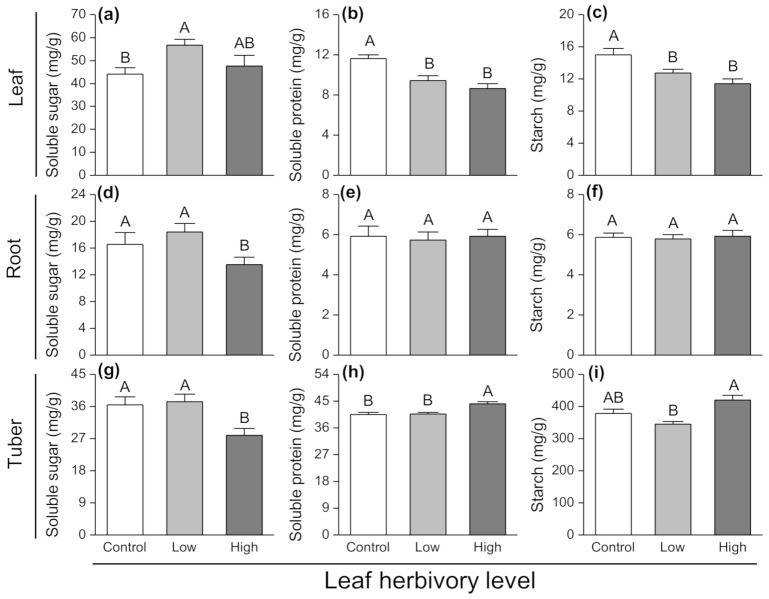
Changes in (**a**–**c**) soluble sugar, (**d**–**f**) soluble protein and (**g**–**i**) starch contents in different plant components (leaf, root and tuber) of potato plants exposed to different herbivory levels of *Phthorimaea operculella*. Data are means ± SE. Different letters indicate significant differences among treatments (*p* < 0.05) based on post hoc Tukey’s HSD test.

**Figure 4 insects-11-00633-f004:**
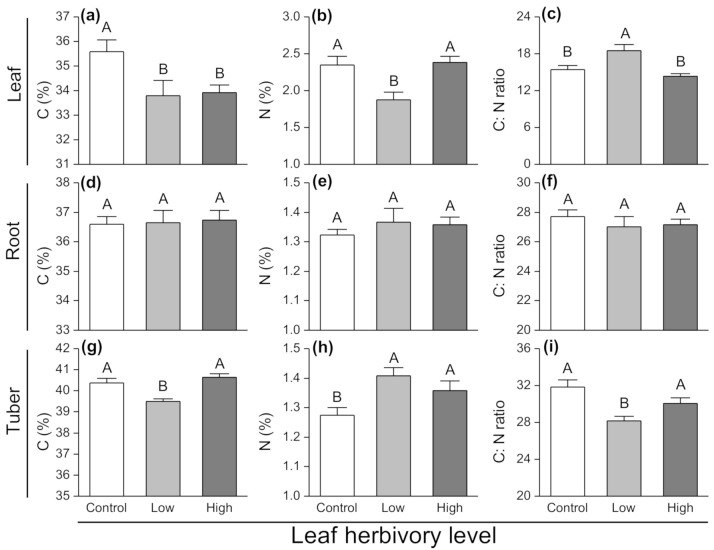
Changes in (**a**–**c**) C content (**d**–**f**), N content and (**g**–**i**) C:N ratio in different plant components (leaf, root and tuber) of potato plants exposed to different herbivory levels of *Phthorimaea operculella*. Data are means ± SE. Different letters indicate significant differences among treatments (*p* < 0.05) based on post hoc Tukey’s HSD test.

**Figure 5 insects-11-00633-f005:**
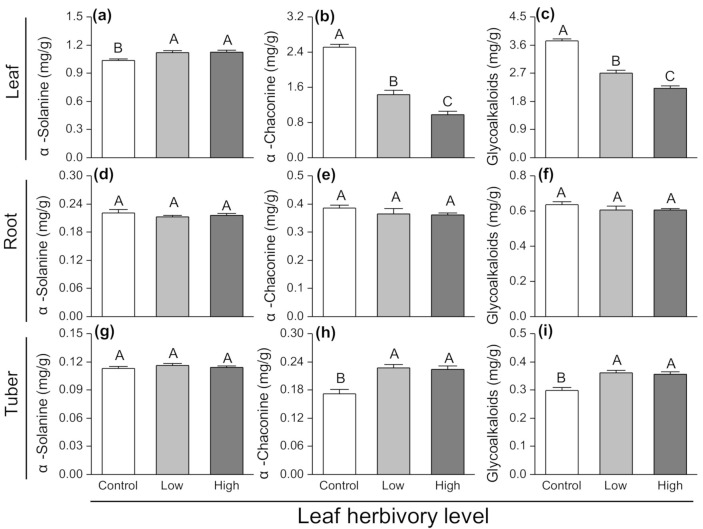
Changes in (**a**–**c**) α-solanine, (**d**–**f**) α-chaconine and (**g**–**i**) glycoalkaloid contents in different plant components (leaf, root and tuber) of potato plants exposed to different herbivory levels of *Phthorimaea operculella*. Data are means ± SE. Different letters indicate significant differences among treatments (*p* < 0.05) based on post hoc Tukey’s HSD test.

**Table 1 insects-11-00633-t001:** Results of two-way ANCOVAs analyses on the effects of aboveground *Phthorimaea operculella* feeding on the larval and pupal developmental times and pupal weight of conspecific larvae of different sex in potato tuber tissues.

Variable	Treatment		Sex		Treatment × Sex	
	F_2,66_	*p*	F_1,67_	*p*	F_5,63_	*p*
Laval developmental time (days)	0.759	0.472	1.882	0.175	3.240	**0.046**
Pupal developmental time (days)	0.800	0.454	0.082	0.776	2.546	0.086
Pupal weight (mg)	1.846	0.166	69.599	**<0.001**	2.986	0.058

(Significant results are in bold).

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
