# Peer review of "Potato Tuberworm Phthorimaea operculella (Zeller) (Lepidoptera: Gelechioidea) Leaf Infestation Affects Performance of Conspecific Larvae on Harvested Tubers by Inducing Chemical Defenses"

_insects, 2020, doi:10.3390/insects11090633_

Round 1
Reviewer 1 Report
The manuscript entitles “Potato Tuberworm (Lepidoptera: Gelechioidea) Leaf Infestation Affects Performance of Conspecific Larvae on Harvested Tubers by Inducing Cecmical Defenses” reports the effects of the damages of potato leaves by potato tuberworm larvae on the performance of the same insect on harvested tubers. To assess the performance the authors measures the larval developmental time, pupal weight and pupal development time of tuberworms reared on previously affected potatoes. Also they measured some nutritional and biochemical parameters of the plants and tubers.
In general, the information is useful and I believe it will contribute to the literature. Overall, I found the MS is written well and organized. My only concern is that, can we conclude such a summary only based on a slight difference in two parameters: larval developmental time of female and pupal weight of male? With the discretion of the editor, the MS may be published in the journal “insects”.
Some specific suggestions:
Lines 95-97: remove from the Materials and Methods, and move to the introduction
Line 102: peat moss (company name, location)
Lines 106-111: remove from the Materials and Methods, and move to the introduction
Line 118: what was to store temperature?
Line 130: what was the greenhouse conditions?
Line 135: envelops (means paper bags?)
Lines 150-151: make it active form, (check the whole section and make the sentences uniform, here mostly active form not passive)
Author Response
Dear Reviewer,
Thanks for your constructive advices on our paper. I have taken into account all your comments and revised these sentences which need to be improved one by one based on your advices. Here, I would like to show our idea and how did we revised by showing the line number in the revised version! Please check them. Thank you very much!
Comment: Can we conclude such a summary only based on a slight difference in two parameters: larval developmental time of female and pupal weight of male?
Response: In previous studies, these developmental parameters (larval developmental time, pupal weight and pupal developmental time of female and male) were often evaluated in potato tuber of different varieties (Golizadeh and Esmaeili 2012; Horgan et al. 2012; Alipour and Mehrkhou 2018). Our study measured how these parameters response to aboveground herbivory, but only two parameters have significant difference. Moreover, we also showed that induced-glycoalkaloids can affect insect performance in discussion (see Line 336-341). Therefore, we think the results can conclude such a summary. Thank you very much for your advice!
Some specific suggestions:
Comment: Lines 95-97: remove from the Materials and Methods, and move to the introduction
Response: We have moved these sentences to Introduction (see Line 95-98 in revised version).
Comment: Line 102: peat moss (company name, location)
Response: We have added company name and location, “peat moss (Jiangsu Beilei Technology Development Co., Ltd. Zhenjiang, Jiangsu, China)” (see Line 133 in revised version).
Comment: Lines 106-111: remove from the Materials and Methods, and move to the introduction
Response: We have moved these sentences to the Introduction (see Line 90-92 and 94-95).
Comment: Line 118: what was to store temperature?
Response: We have provided the store conditions, “in above laboratory conditions (27±2℃, 70-80% R.H. and a 12:12 h L: D photoperiod)” (see Line 151-152, in revised version).
Comment: Line 130: what was the greenhouse conditions?
Response: We have provided the greenhouse conditions, “in the greenhouse with natural temperature and light.” (see Line 165, in revised version).
Comment: Line 135: envelops (means paper bags?)
Response: ‘envelops’ means ‘paper bags’. We have changed ‘envelops’ to ‘paper bags’ (see Line 171, in revised version).
Comment: Line 150-151: make it active form, (check the whole section and make the sentences uniform, here mostly active form not passive)
Response: We have revised all active forms to make the sentences uniform, (see Line 185-190).
Reviewer 2 Report
This paper gives important information about the effect of a leaf feeder on the potato plant and its effect on conspecific individuals feeding on the roots. Such studies on the insect impact on the plant are always important and sometimes underestimated. However, a careful revision of the manuscript is highly recommended before reconsider it for publication. Introduction needs a strong revision: some parts are too long, and often not accurate. Materials and methods are not always clear and should be improved. Results are correctly analyzed and fairly reported. Discussion of the results could be improved, underlying why the results obtained here are important and which consequences can have also for future perspectives. Overall the paper needs strong revision of English by a native speaker, many sentences are unclear, particularly in the abstract and in the introduction.
Line 13-14 as opening sentence it should be written better, suggest revising. Also consider that in this study there is a unique species, while from the first sentence I was understanding the presence at least of two different ones (one leaf feeder and one root feeder)
Line 15-18, please rewrite.
Line 22-24 why this parameters are important? Please give some brief detail
Line 37-38 if there are “many” I suggest not give just the example of the “plant type” (do the authors mean plant species??? Or…?)
Line 51-69, I think this paragraph is too long and need to be summarized.
Line 56-57 and across the manuscript, when a species is reported for the first time please give the detail of the order and family and also the descriptor.
Line 57-58 this sentence should be written better.
Line 86-91 this part is unclear. Which sentence refers to the objective 1 and objective 2? It needs to be strongly revised, moreover I suggest to avoid such type of expressions as “We conducted…”, “We measured…”, “We predicted…”
Line 95-97, please delete, these are not MM. Just report how potatoes were grown.
Line 95-104 important info are missing: mean temperature, relative humidity, photoperiod.
Line 106-111 already reported before, and again this is not a MM. This part should start as “A colony of P. operculella was started from….”
line 120. why write Experiment1 and Experiment2? moreover the titles of the different experiments reported in mm should be the same in the results, otherwise this part could be misleading.
line 124-125, here there are lots of run-on stentences, plese reformulate.
Line 128-130 was this done also for the control plant? The nylon net can influence the photosynthesis rate.
Line 138, how was the chemical composition observed?
Line 139, why don’t the authors use a larger pot? please rewrite, or delete.
Line 149 check misspellings
Line 150-151, “using the methodology suggested by…”
Line 198, many information of the caption are redundant, once already reported in the text, same for all the other figures captions.
line 324-325, this sentence should be placed in the right context.
Line 337 check
Line 343-346, this part should be rewritten highlighting more in detail the consequences of these findings.
Author Response
Dear Reviewer,
Thanks for your advice on my paper.I have taken into account all your comments and revised these figures which need to be improved one by one based on your advices. Here, I would like to show our idea and how did we revised by showing the line number in the revised version without track change! Please check them. Thank you very much!
Comment: Line 13-14 as opening sentence it should be written better, suggest revising. Also consider that in this study there is a unique species, while from the first sentence I was understanding the presence at least of two different ones (one leaf feeder and one root feeder)
Response: Thanks to your advices. We have revised this sentence by changing ‘leaf and root feeders’ to ‘aboveground and belowground herbivores’ (see Line 24 in revised version).
Comment: Line 15-18, please rewrite.
Response: We think it is better to delete it than rewrite it. The first sentence of “abstract” should be followed by a direct description of what we should do, we have deleted this sentence (see Line 26-30 in revised version).
Comment: Line 37-38 if there are “many” I suggest not give just the example of the “plant type” (do the authors mean plant species??? Or…?)
Response: We mean plant species, we also have added a factor ‘herbivore species’, “such as plant species, herbivore species” (see Line 50-51 in revised version).
Comment: Line 51-69, I think this paragraph is too long and need to be summarized.
Response: This paragraph has been summarized. First, we have combined the first three sentences into one. “When plants are subjected to herbivore attacks, secondary metabolites which include glucosinolates, phenols, tannins, glycoalkaloids and terpenoids would significantly change in different plant compartments and then affect insect performance”. Second, we also have combined sentences 7, 8 and 9 into one. “Furthermore, herbivory-induced changes in primary metabolites that involved in the function, growth and life history of different plant organs closely related to herbivore performance” (see Line 64-69) and (Line 77-87 in revised version).
Comment: Line 56-57 and across the manuscript, when a species is reported for the first time please give the detail of the order and family and also the descriptor.
Response: We have added the order and family of species in the manuscript, “Manduca sexta (Lepidoptera: Sphingidae)” “Agriotes lineates (Coleoptera: Elataridae)” “Spodoptera exigua (Lepidoptera: Noctuidae)” “Bikasha collaris (Coleoptera: Chrysomelidae)” “Tecia solanivora (Povolný) (Lepidoptera: Gelechioidea)” “Spodoptera frugiperda (Lepidoptera: Noctuidae)” (see Line 56,71,72,73,74,75, in revised version).
Comment: Line 57-58 this sentence should be written better.
Response: The sentence has been revised, “Above- and belowground herbivory decreased the survival of adult Bikasha collaris (Coleoptera: Chrysomelidae) by increased tannin level” (see Line 72-74 in revised version).
Comment: Line 86-91 this part is unclear. Which sentence refers to the objective 1 and objective 2? It needs to be strongly revised, moreover I suggest to avoid such type of expressions as “We conducted…”, “We measured…”, “We predicted…”
Response: The parts have been revised. We added our aims and rewrote this part (see Line 108-116 in revised version).
Comment: Line 95-97, please delete, these are not MM. Just report how potatoes were grown
Response: We think this part can be moved to Introduction, and another reviewer also suggested that we move this sentence into the Introduction. Thus, we have moved to the Introduction (see 95-98 in revised version).
Comment: Line 95-104 important info are missing: mean temperature, relative humidity, photoperiod.
Response: We have added the conditions. It now reads as “in an open-sides greenhouse with natural temperature and light” (see Line 134-135 in revised version).
Comment: Line 106-111 already reported before, and again this is not a MM. This part should start as “A colony of P. operculella was started from….”
Response: We think this part can be moved to Introduction, and another reviewer suggested that we move this sentence into the Introduction. Thus, we have moved to the Introduction (see Line 90-92 and 94-95).
Comment: Line 120. why write Experiment1 and Experiment2? moreover the titles of the different experiments reported in mm should be the same in the results, otherwise this part could be misleading.
Response: At first, we ran two experiments, so wrote Experiment 1 and Experiment 2. We have deleted them to avoid misleading (see Line 154 and Line 177).
Comment: Line 124-125, here there are lots of run-on sentences, please reformulate
Response: These sentences have been revised, “(1) control (0 larvae per plant), (2) low herbivory (9 larvae per plant) and (3) high herbivory (18 larvae per plant), and each level had 20 replicates.” (see Line 158-159).
Comment: Line 128-130 was this done also for the control plant? The nylon net can influence the photosynthesis rate.
Response: Yes, the control plants were also covered with net bags. We have added and revised, “Control plants were also covered with small net bags but did not receive any larvae.” (see Line 163).
Comment: Line 138, how was the chemical composition observed?
Response: What we want to express is to prepare for chemical analysis, we have revised by changing “to determine their chemical compositions” to “for chemical analysis” (see Line 173).
Comment: Line 139, why don’t the authors use a larger pot? please rewrite, or delete.
Response: Before the experiment, we thought this type of pot (height: 25 cm, diameter: 30 cm) was big enough, but it was still small. We have deleted the sentence as suggested (see Line 174-176).
Comment: Line 149 check misspellings
Response: We think there is something wrong with the use of the word ‘held’, we have changed ‘held’ to ‘put’ (see Line 186).
Comment: Line 150-151, “using the methodology suggested by…”
Response: We have revised as suggested by changing “by methodology of Rondon” to “using the methodology suggested by Rondon” (see Line 188).
Comment: Line 198, many information of the caption are redundant, once already reported in the text, same for all the other figures captions.
Response: We have deleted partial repetition, but we also appropriately retained some, for example, “Data are means ± SE. Different letters indicate significant differences among treatments (P <0.05) based on post hoc Tukey’s HSD test.” (See Line 262-264, Line 285-287, Line 302-304 and Line 321-323).
Comment: Line 324-325, this sentence should be placed in the right context.
Response: Here, our results and previous study are similar, indicating that there are obvious differences in the developmental performance of male and female. So we showed our results first, and then showed that previous studies are very similar to our results (see Line 366-367).
Comment: Line 337 check
Response: We have checked and deleted extra character (see Line 377).
Comment: Line 343-346, this part should be rewritten highlighting more in detail the consequences of these findings.
Response: The part has been rewritten and added these detail of our findings. “Our results suggest that either low or high herbivory by P. operculella affected concentration of tubers chemicals (i.e. soluble sugar, soluble protein, starch, C, N, C: N ratio, α-chaconine, α-solanine and glycoalkaloids). Similarly, chemical levels in the leaves were also affected, except for α-solanine. However, in roots, only high herbivory reduced soluble sugar levels. These significant changes in phytochemical reallocation between leaves and tubers in responses to aboveground herbivory may indicate an intimate connection between plant-herbivore interaction and plant physiology.” (see Line 384-389).
Round 2
Reviewer 2 Report
Line 24. I continue to think that a sentence like “Conspecific aboveground and belowground herbivores can interact each other…” should be better”. Otherwise I continue to think that you are talking about different species, one feeding on roots and the other from leaves.
Line 98-100. I think that authors should refer only to the species that is the objective of this study, instead of “leaf and tuber herbivores”, in fact the study is focused on conspecific individuals feeding on leaves or roots
Line 131. Please change “did not receive any larvae..” in “were not infested”
Line 318. P. operculella goes in italic.
Author Response
Dear Reviewer,
Thanks for your advice for our manuscript. We have carefully considered these comments. For details please see below our response which shows how we addressed the issues point by point.
Comment: Line 24. I continue to think that a sentence like “Conspecific aboveground and belowground herbivores can interact each other…” should be better”. Otherwise I continue to think that you are talking about different species, one feeding on roots and the other from leaves.
Response: We have revised this as suggested. (see Line 24 in revised version)
Comment: Line 98-100. I think that authors should refer only to the species that is the objective of this study, instead of “leaf and tuber herbivores”, in fact the study is focused on conspecific individuals feeding on leaves or roots.
Response: We have revised this by changing “leaf and tuber herbivores” to “of P. operculella larvae feeding leaves and tubers”. (see Line 101 in revised version)
Comment: Line 131. Please change “did not receive any larvae.” in “were not infested”
Response: We have revised this as suggested. (see Line 132 in revised version)
Comment: Line 318. P. operculella goes in italic.
Response: We have revised this as suggested. (see Line 319 in revised version)